# Secondary Malignancies after Ewing Sarcoma—Epidemiological and Clinical Analysis of an International Trial Registry

**DOI:** 10.3390/cancers14235920

**Published:** 2022-11-30

**Authors:** Isabelle Kaiser, Katja Kauertz, Stefan K. Zöllner, Wolfgang Hartmann, Thorsten Langer, Heribert Jürgens, Andreas Ranft, Uta Dirksen

**Affiliations:** 1Pediatrics III, University Hospital Essen, 45147 Essen, Germany; 2West German Cancer Center (WTZ), University Hospital Essen, 45147 Essen, Germany; 3German Cancer Consortium (DKTK), Essen/Düsseldorf, University Hospital Essen, 45147 Essen, Germany; 4Gerhard Domagk Institute for Pathology, University Hospital Muenster, 48149 Muenster, Germany; 5Pediatric Hematology and Oncology, LESS Group, University Medical Center Schleswig-Holstein, 23538 Luebeck, Germany; 6Pediatric Hematology and Oncology, University Children’s Hospital Münster, 48149 Münster, Germany

**Keywords:** childhood cancer, cancer survivor, Ewing sarcoma, secondary malignant neoplasms, secondary malignancy, radiotherapy

## Abstract

**Simple Summary:**

Ewing sarcoma (EwS) is a malignant bone and soft tissue cancer that requires intensive treatment with multiple chemotherapies and either surgery, irradiation, or both as local therapy. For most survivors of EwS, long-term sequelae such as secondary malignant neoplasms (SMNs) other than EwS are concerning. Few studies suggest that SMNs after EwS are a rare but serious event. Comprehensive data are lacking. We reviewed consecutive EwS trials from the Cooperative Ewing Sarcoma Study (CESS) group to evaluate the features of SMNs in EwS patients. Our analysis revealed 101 cases of SMNs in 96 EwS patients. Solid SMNs were detected more frequently than hematologic SMNs, in 55.2% versus 44.8%. The latency between EwS diagnosis and SMN occurrence was longer for solid SMNs (median: 8.4 years) than for hematologic SMNs (median: 2.4 years) (*p* < 0.001). The survival rate after SMNs was 0.49, with solid SMNs having a significantly better prognosis. Our results confirm the need for a structured follow-up system.

**Abstract:**

Ewing sarcoma (EwS) represents highly aggressive bone and soft tissue tumors that require intensive treatment by multi-chemotherapy, surgery and/or radiotherapy. While therapeutic regimens have increased survival rates, EwS survivors face long-term sequelae that include secondary malignant neoplasms (SMNs). Consequently, more knowledge about EwS patients who develop SMNs is needed to identify high-risk patients and adjust follow-up strategies. We retrospectively analyzed data from 4518 EwS patients treated in five consecutive EwS trials from the Cooperative Ewing Sarcoma Study (CESS) group. Ninety-six patients developed SMNs after primary EwS, including 53 (55.2%) with solid tumors. The latency period between EwS and the first SMN was significantly longer for the development of solid SMNs (median: 8.4 years) than for hematologic SMNs (median: 2.4 years) (*p* < 0.001). The cumulative incidence (CI) of SMNs in general increased over time from 0.04 at 10 years to 0.14 at 30 years; notably, the specific CI for hematologic SMNs remained stable over the different decades, whereas for solid SMNs it gradually increased over time and was higher for metastatic patients than in localized EwS patients (20 years: 0.14 vs. 0.06; *p* < 0.01). The clinical characteristics of primary EwS did not differ between patients with or without SMNs. All EwS patients received multi-chemotherapy with adjuvant radiotherapy in 77 of 96 (80.2%) patients, and the use of radiation doses ≥ 60 Gy correlated with the occurrence of SMNs. The survival rate after SMNs was 0.49, with a significantly better outcome for solid SMNs compared with hematologic SMNs (3 years: 0.70 vs. 0.24, respectively; *p* < 0.001). The occurrence of SMNs after EwS remains a rare event but requires a structured follow-up system because it is associated with high morbidity and mortality.

## 1. Introduction

Ewing sarcoma (EwS) is a rare, highly aggressive malignancy of small blue, round cells that occurs in bone and soft tissue and predominantly affects children, adolescents and young adults [1]. An incidence of 1.5 cases per million is observed in people of European descent [1,2,3]. Twenty to 25% of patients have metastases at the time of initial diagnosis, which is the most important prognostic factor for EwS [4].

Although 85% of all cases have a balanced chromosomal translocation in which Ewing sarcoma breakpoint region 1 (EWSR1) protein fuses with the Friend leukemia integration 1 (FLI1) transcription factor, resulting in the *EWSR1-FLI1* fusion oncogene [5,6,7], current first-line treatment of EwS does not include targeted therapy. EwS requires a multimodal therapeutic approach consisting of multiple cycles of multiagent chemotherapy and local therapy consisting of surgery, radiotherapy, or both modalities. The decision on local treatment can be discussed in specialized tumor boards [8]. Rational combinatorial and dose-intensifying strategies of the basic chemotherapeutic regimen have improved outcome in all patients except those with disseminated disease [9,10,11].

As a consequence of improved survival, long-term surveillance for late effects has become increasingly important because the occurrence of secondary malignant neoplasms (SMNs) causes high morbidity and mortality [12,13]. SMNs are defined as malignancies that occur during or after cancer treatment and are not detected before the initial cancer treatment. Histologically, SMNs present as distinct from the primary tumor [14].

Few reports of SMNs after EwS have been published, mostly in association with trial reports. Nevertheless, important aspects such as the clinical characteristics of EwS patients who develop SMNs, the types of SMNs, and the outcomes for patients with SMNs after EwS have not yet been adequately studied.

Several risk factors have been associated with SMNs [12,14]. Previous studies showed a leukemogenic potential of epipodophyllotoxins, anthracyclines, and alkylating agents [15,16,17,18,19,20,21] and an increased risk of solid SMNs after high-dose radiotherapy (≥60 Gy) [22,23,24,25].

In the present study, we retrospectively investigated the epidemiology and clinical features of 96 identified patients with SMNs in an international cohort of 4518 patients treated for EwS in five consecutive clinical trials. Specifically, we aimed to identify specific clinical features in EwS patients and associated treatments that might predispose to SMNs. 

## 2. Materials and Methods

### 2.1. Patient Cohorts and Eligibility Criteria

This retrospective analysis included data from 4518 international patients treated for EwS between 1980 and 2019 and registered in the Ewing Sarcoma Study Group of the GPOH (German Society of Pediatric Hematology and Oncology) database. The corresponding phase III clinical trials were Cooperative Ewing’s Sarcoma Studies 1981 (CESS 81, 184 patients), Cooperative Ewing’s Sarcoma Studies 1986 (CESS 86, 490 patients), European Intergroup Cooperative Ewing’s Sarcoma Study 92 (EICESS 92, 875 patients), EUROpean Ewing tumor Working Initiative of National Groups—Ewing Tumor Studies-1999 (Euro-E.W.I.N.G. 99, 1548 patients), and Ewing 2008 (1421 patients). 

Phase III randomized clinical trials were multicenter and nationwide or international in scope. Informed consent for long-term follow-up and data analysis was obtained from patients, parents, or guardians. Ethics committee approval was obtained at baseline. All trials are summarized in Table 1 and briefly described below.

In the CESS 81 trial (patient enrollment from 1981 to 1985), four cycles (nine weeks each) of VACA chemotherapy were administered, which included vincristine, actinomycin D, cyclophosphamide, and adriamycin. Two cycles of VACA were followed by local therapy consisting of either surgery, surgery plus postoperative radiotherapy (36 Gy), or definitive radiotherapy (46 Gy or 60 Gy, 36 Gy to the entire bone) [22,26].

In the CESS 86 trial (patient enrollment from 1986 to 1991), patients were divided into two risk groups according to tumor volume, tumor location, and metastatic status. Standard-risk (SR) patients had small (<100 mL) extremity tumors and were treated with VACA. If an initial tumor volume ≥100 mL, central tumor localization, or metastatic disease was diagnosed, patients were classified as high-risk (HR) and received a VAIA regimen (vincristine, actinomycin D, ifosfamide, adriamycin). Local therapy was given after one cycle of VAIA (week 9/10) and included surgery, postoperative radiotherapy (44.8/45 Gy), or definitive radiotherapy (44.8/45 Gy with a local boost up to 60 Gy). Patients received a total of four chemotherapy cycles with twelve courses [22,26,27]. 

In the EICESS 92 trial (patient enrollment from 1992 to 1999), patients were divided into SR and HR groups according to tumor volume and metastatic status, with a threshold of 100 mL, and chemotherapy doses were changed. SR patients (tumor volume < 100 mL) were randomly assigned to receive four courses of VAIA followed by either ten courses of VAIA or VACA. HR patients (≥100 mL and/or metastatic disease) received VAIA or VAIA plus etoposide (EVAIA) over 14 courses. Preoperative radiotherapy (45–55 Gy depending on the anticipated extent of resection) was initiated at EICESS 92. It was applied after the sixth week of chemotherapy if the soft tissue component had decreased by less than 50% after the second chemotherapy or if the tumor was deemed unresectable.

Definitive radiotherapy (up to 55 Gy) was administered after the fourth course. Patients with wide resection but poor histological response (≥10% viable tumor) or marginal resection but good response (<10% viable tumor) received postoperative radiotherapy at a dose of 45 Gy. For intralesional resection or marginal resection and poor histological response, up to 55 Gy was recommended. In patients with pulmonary metastases, the entire lung was irradiated (15 Gy if < 14 years and 18 Gy if ≥ 14 years) [26,28].

In the Euro-E.W.I.N.G. 99 trial (patient enrollment from 1999 to 2011), patients were stratified into three risk groups. All patients received six induction courses of VIDE (vincristine, ifosfamide, doxorubicin, etoposide) and one course of VAI (vincristine, actinomycin D, ifosfamide). The R1 patients with localized disease and either a favorable response to neoadjuvant chemotherapy (<10% viable tumor) or a small initial tumor volume (<200 mL) were randomly assigned to either adjuvant therapy with vincristine, actinomycin D, and cyclophosphamide (VAC) or vincristine, actinomycin D, and ifosfamide (VAI) [29]. R2loc patients with localized disease and either poor histological response to neoadjuvant chemotherapy (≥10% viable tumor) or a large initial tumor volume (≥200 mL) and the R2pulm patients with isolated lung metastases were randomized to receive seven courses of VAI versus high-dose chemotherapy with busulfan-melphalan followed by retransfusion of autologous hematopoietic stem cells [30]. R2pulm patients randomized to VAI also received whole-lung irradiation [31]. R3 patients with disseminated disease were enrolled in a non-randomized arm and were scheduled to receive high-dose chemotherapy [11]. In the Euro-E.W.I.N.G. 99 trial, definitive surgical resection was recommended whenever possible and postoperative radiotherapy for large primary tumors, unfavorable histological response, and marginal resection; preoperative radiotherapy was optional. Definitive radiotherapy was considered for unresectable tumors. The following doses were recommended for radiotherapy: preoperative radiotherapy 54.4 Gy, definitive radiotherapy 44.8 Gy (with a boost of 54 Gy, but in individual cases with a maximum boost of 64 Gy depending on tumor location and patients age), and postoperative radiotherapy 44.8 Gy–54.4 Gy [32]. 

The Ewing 2008 trial (patient enrollment from 2008 to 2019) was the follow-up trial of Euro E.W.I.N.G. 99 and stratified patients accordingly but asked randomized questions in all risk groups. All patients received the VIDE induction regimen. Standard-risk patients (tumors < 200 mL and/or favorable histologic response to induction chemotherapy) received sex-specific maintenance therapy with VAC in women and VAI in men, and patients were randomized to the addition of zolendronic acid or not. The R2 arms were adopted from the Euro E.W.I.N.G. 99 trial [30,31]. Patients with disseminated disease were defined as very high-risk (R3) and received either eight cycles of VAC or eight cycles of VAC plus a course of high-dose treosulfan-melphalan chemotherapy (followed by autologous stem cell reinfusion) after induction chemotherapy. Local therapy was delivered as surgery, whenever possible, and/or radiotherapy [30,31]. 

### 2.2. Follow-Up and Statistical Analysis 

Cases of SMNs (secondary malignant neoplasm defined as cancer of a histological type other than EwS that occurs during or after cancer treatment and was not detected before the initial cancer treatment) were reported by participating institutions and confirmed by pathology report. Survival after SMNs was estimated using the Kaplan–Meier method and defined as the interval between the diagnosis of SMNs and the date of death or last contact. Cumulative incidences of SMNs were estimated with XLSTAT using competing risk analysis. SAS and SPSS were used for exploratory data analysis. 

### 2.3. Literature Search

A Medline search of the PubMed database was performed using the following terms: “second malignant neoplasm,” “SMN,” “secondary cancer,” “second malignancy”, and “childhood cancer survivor”. 

## 3. Results 

### 3.1. Patient Characteristics and Clinical Features of Primary Ewing Sarcoma

Of the 4518 EwS patients treated between 1980 and 2019 according to EwS study protocols, 101 cases of subsequent malignant neoplasms were retrospectively assigned to 96 patients. Five patients presented with two different malignant neoplasms after the primary diagnosis of EwS. The epidemiological and clinical characteristics of EwS patients with SMNs are summarized in Table 2. The median time from diagnosis of SMNs to last follow-up was 1.16 years (range: 0–19). The median time from diagnosis of EwS to diagnosis of SMNs was 4.9 years (0.1–28.1). Four patients were lost to follow-up. Loss to follow-up resulted from relocation, change of oncologist, or refusal of further contact for follow-up. The clinical characteristics of the five EwS patients with two SMNs are summarized in Appendix A. In the following, only the results of patients with a single SMN are listed: 51 of the 96 patients with SMNs were female, and 45 were male. The mean age of EwS patients at diagnosis was 14.4 years and ranged from 2.4 to 68.6 years. Thirty-one patients (32.3%) had metastases at the time of EwS diagnosis. In most patients, metastases were located to the lung (18.6%), bone marrow (7.2%), and bone (13.4%), but not at the site of the primary tumor. None of the patients with SMNs had a reported family history of cancer predisposition or tumor-associated syndrome.

In our study cohort, all patients received chemotherapy. Table 1 provides an overview of the type and dosage of chemotherapy. In all studies, systemic treatment was supplemented by local therapy using surgery and/or radiotherapy. Radiotherapy was given either as definitive therapy or as a local adjuvant approach after surgery. 77 of 96 patients (80.2%) received radiotherapy. In the CESS 86 study, 11 of 16 SMNs patients were treated with high-dose radiotherapy (≥60 Gy). These patients suffered predominantly from solid tumors (*n* = 9) such as sarcomas including osteosarcomas (*n* = 5) or carcinomas (*n* = 4). 

### 3.2. Epidemiology of Secondary Malignant Neoplasms

The types of SMNs after primary EwS are summarized in Table 3 according to the EwS study protocols. The corresponding data on EwS patients with more than one SMN are summarized in Appendix A. The highest percentage of patients with SMNs was found in the CESS 86 study (3.1%), and the lowest percentage of SMNs was documented in patients in the CESS 81 study (1.1%). Solid tumors were detected more frequently than hematologic neoplasms after primary EwS, in 55.2% versus 44.8%. Carcinomas formed the largest group (*n* = 23) of all solid tumors. Among sarcomas, osteosarcoma was the most common type of sarcoma, followed by rhabdomyosarcoma. The most common hematologic neoplasms were acute myeloid leukemia (AML) and myelodysplastic syndrome (MDS). In two cases, incomplete documentation prevented clear classification between MDS and AML. Other SMNs included astrocytoma (*n* = 1), blastoma (*n* = 1), melanoma (*n* = 2), neuroendocrine tumor (*n* = 1), and pancreatic tumor (*n* = 1). The median time between EwS and SMNs was 5.4 years. The latency period was longer for solid tumors (median: 8.4 years) than for hematologic neoplasms (median: 2.4 years) (d = 1.22; *p* < 0.001). The median time between EwS and tertiary malignant neoplasm was 9.4 years, and the median time between SMNs and tertiary malignant neoplasm was 1.4 years.

### 3.3. Cumulative Incidences and Outcome of Secondary Malignant Neoplasms

The cumulative incidence (CI) of SMNs was 0.04 (SE < 0.01) at 10 years, 0.07 (SE = 0.01) at 20 years, and 0.14 (SE = 0.03) at 30 years. For solid tumors, the specific CI was 0.02 (SE < 0.01) at 10 years, 0.06 (SE = 0.01) at 20 years, and 0.12 (SE = 0.03) at 30 years. For hematologic neoplasms, the specific CI was 0.02 (SE < 0.01) after each of 10, 20 and 30 years. For hematologic neoplasms, the specific CI reached a plateau after 8 years. Female patients had a higher risk than male patients (20 years: 0.11 vs. 0.05; *p* = 0.03), similarly for metastatic patients compared to localized patients (20 years: 0.14 vs. 0.06; *p* < 0.01). Age (</≥15 years) had no effect on cumulative incidence. The type of local treatment did not affect the incidence of SMNs in general, but the use of radiation doses ≥ 60 Gy correlated with the incidence of SMNs in particular. The survival rate after SMNs was 0.49 (SE = 0.06). It differed significantly between solid and hematologic SMNs for EwS patients (3 years: 0.70 vs. 0.24; *p* < 0.001) (Figure 1). 

### 3.4. Literature Search

We identified the seven specific articles presented in Table 4 to compare and discuss risk-associated factors for SMNs after EwS. The articles included studies focused on the development of SMNs in patients with EwS or Ewing sarcoma family tumors. 55.9% of patients were female (4/7 articles) in a median cohort size of 381 EwS patients (*n* = 7/7). The median age at the time of EwS diagnosis was 14.7 years (*n* = 5/7) and the latency period was 8.2 years (*n* = 6/7). Cumulative incidences ranged from 4.7% at 10 years to 15% at 25 years. A solid tumor was identified as SMNs in 65.5% (*n* = 6/7). The tumors most frequently described in the articles were breast cancer, sarcoma including osteosarcoma, MDS, and leukemia including AML.

## 4. Discussion

A 2017 report from the German Childhood Cancer Registry showed that the 15-year survival rates for childhood cancer patients younger than 15 years have increased to 82% [38]. Refined treatment protocols for EwS aim to reduce therapy-related toxic effects, resulting in higher survival rates and longer follow-up [4,34]. However, one of the most devastating complications of primary cancer therapy is the development of a second malignancy.

The most recent report (2019) from the German Childhood Cancer Registry published a cumulative incidence of 6.8% for SMN in German cancer survivors (diagnosed < 18 years) within 30 years of diagnosis. Focusing on SMNs after EwS, the registry reported a cumulative incidence of 4.4% at 30 years (treated between 1981 and 2016) [39]. In contrast, a 2020 analysis of the Surveillance, Epidemiology and End Results Program (SEER) database by Friedman et al. published a cumulative incidence of 10.1% at 30 years (treated between 1970 and 1986) [40].

Our study reports comprehensive epidemiologic and clinical data on SMNs after primary EwS in a series of international EwS trials spanning 30 years of patient enrollment and standardized treatment, making it the largest and longest retrospective study of primary EwS patients with SMNs.

We determined cumulative incidences (CI) of 1.9%, 3.9%, 5%, 7.4% and 14% at 5, 10, 15, 20, and 30 years, respectively. The CI of SMNs in patients treated for EwS varies widely in the literature (Table 4). In a meta-analysis, Caruso et al. reported a CI ranging from less than 0.9% at 5 years to more than 20.5% at 30 years [12]. Friedman et al. (Memorial Sloan Kettering) described a CI of 15% at 25 years despite their more recent publication date, but this may be due to a long observation period of 38 years starting in 1974 [37]. Longhi et al. (Italian Sarcoma group) reported a CI of only 5% at 25 years but excluded patients with disseminated disease and thus those who received extensive treatment [36]. In the Children’s Oncology Group’s recent trial of the treatment of localized EwS (AEWS1031), patients were randomly assigned to two different regimens. In regimen A, patients received 17 cycles of compressed chemotherapy with a standard five-drug interval, while patients in regimen B received experimental therapy with five cycles of vincristine, topotecan, and cyclophosphamide within the 17 cycles. A numerical rate of 4.3% for SMNs was reported in a total of 626 patients. A cumulative incidence for comparison was not reported, nor was information on SMNs characteristics (e.g., tumor entity). Leavey et al. did not detect a significant difference in SMNs between the randomized study arms and hypothesized that vincristine, topotecan and cyclophosphamide had no effect on the risk of SMNs [41].

The median latency from primary EwS diagnosis to SMNs in our analysis was 5.4 years. Similar results were published by Kuttesch et al. (7.6 years) and are consistent with reported latencies of SMNs after other childhood tumors [24,37,42]. In our cohort, latency was related to the type of SMNs, with hematologic neoplasms (median 2.4 years) occurring earlier after primary EwS than solid tumors (median 8.4 years). The study by Friedman et al. reported a comparable median latency of 3.2 years for secondary AML/MDS. For solid tumors, the latency period in this study was 21.3 years, twice as long as our results. It remains unclear whether the above discrepancy is due to the limited number of only 300 patients observed by Friedman or to the non-Ewing round cell tumors included [37]. However, we suspect that the incidence of solid malignancies in our cohort may be underestimated because we included data from more recent trials with limited follow-up. In other studies, the incidence of hematologic neoplasms may be underestimated because the studies by Kuttesch et al. and Ginsberg et al. excluded patients who died within the first three and five years after diagnosis, respectively [24,34].

Our analysis shows that the risk of solid SMNs does not plateau but increases over time. Solid tumors commonly include carcinomas and osteosarcomas, and they can occur up to 28 years after the primary EwS diagnosis.

Several studies have found breast cancer to be the most common SMN in childhood cancer survivors [43,44,45,46]. A high incidence of breast cancer has been described especially after chest irradiation in Hodgkin’s lymphoma and after total lung irradiation for EwS patients [43,44,45]. We observed a remarkably low number of breast cancer cases in our cohort (three cases exclusively in women). All patients received radiotherapy to the thoracic wall and developed tumors within the irradiation field. In the study by Schellong et al., 26 cases of breast cancer were detected in 590 Hodgkin’s lymphoma patients, of which 25 tumors were located in the previously irradiated field, although the radiation doses for the supradiaphragmatic fields ranged only from 20 to 45 Gy [47]. In our patients, EwS was diagnosed at a median age of 15.6 years (range: 12–21 years), supporting the observations of Bhatia et al. who suggested an association between radiation exposure between the ages of 10 and 16 years and the development of breast cancer as SMNs. This could be caused by exposure of carcinogens in the growing breast tissue as shown by previous data [46]. We could not conclusively explain the low number of breast cancer cases detected in our cohort.

Several studies have confirmed initial status of EwS disease, young age at initial diagnosis (<10 years or <14 years), female gender, radiotherapy, alkylating agents as well as topoisomerase inhibitors, high-dose chemotherapy followed by autologous stem cell transplantation including granulocyte colony-stimulating factor (G-CSF), and tumor predisposition syndromes as risk factors for SMNs in EwS patients [12,24,48,49,50]. The median age at EwS diagnosis in our cohort was 16 years, which is consistent with the overall median age at EwS diagnosis (15 years) and previous reports of EwS with SMNs with a median age of 14.8 years (range: 0–40 years) (Table 4) [1]. While Navid et al. did not find an age-related risk, Kaatsch et al. showed that children younger than 10 years with EwS treated with radiotherapy were at higher risk for developing SMNs [35,51]. EwS patients with SMNs showed an inverse gender distribution compared with the typical distribution in EwS patients of 40% females and 60% males [3]. Female gender has also been associated with an increased risk of SMNs in other studies looking at the development of SMNs in specific childhood EwS and other cancer patients [27,43]. The long-term effects of radiation therapy on developing tissues remain uncertain, but tissue susceptibility to mutagenic effects may be higher in younger children [51]. Because there is a lack of studies on radiation therapy in children and young adults, the safety limits of radiation doses are unknown [52]. Radiation therapy is highly associated with the development of sarcomas of bone and soft tissue, especially osteosarcomas [53]. Among childhood cancers, EwS is treated with one of the highest doses of radiation [23]. Kuttesch et al. found that EwS patients who received radiation doses of > 48 Gy developed secondary sarcomas but not hematologic neoplasms. The highest risk was found in the group of patients irradiated with ≥ 60 Gy. All secondary sarcomas were located in or in close proximity to the primary irradiation field. The high percentage (92%) of patients treated with radiotherapy may also explain the high number of solid SMNs and the corresponding high cumulative incidences of SMNs [24]. In the study of Dunst et al., even no secondary sarcomas were detected in patients without prior radiotherapy [22]. In our study, patients who developed solid SMNs received higher doses of radiotherapy (mean 53.1 Gy) than hematologic SMNs (mean 49.1 Gy). Most EwS patients with SMNs in the CESS 86 trial were treated with high-dose radiotherapy (≥60 Gy). Solid SMNs were more likely to occur, of which > 50% were localized in the former irradiation area. Osteosarcomas as SMNs seemed to have reduced over time. We hypothesize that the risk decreased over time due to the amelioration in radiotherapy regimen (for example the improvement of irradiation plans as well as techniques). In general, the initiation of preoperative radiotherapy in the EICESS 92 may have also opened the option of surgery (instead of definitive radiotherapy) for tumors deemed as unresectable before and therefore reduced the applied dosage. Previous publications also assumed a risk association of radiotherapy with concomitant use of alkylating agents [24,25,33]. In previous studies of cancer survivors, alkylating agents, topoisomerase inhibitors (epipodophyllotoxins, e.g., etoposide), and the adjunctive use of anthracyclines [54] have been associated with the development of SMNs and may cause stem cell damage predisposing to secondary myelodysplastic syndrome and leukemia [15,17,18,19,20,21,48,55,56,57]. In addition to leukemic potential, exposure to anthracycline and alkylating agents has been reported to increase the risk of subsequent solid neoplasms [19,21,58,59]. Anthracyclines, topoisomerase inhibitors, and alkylating agents have been used extensively in EwS treatment (Table 1) [60]. While the number of cycles increased, the cumulative doses of most chemotherapeutic agents (vincristine, cyclophosphamide, doxorubicin, and etoposide) gradually decreased over the course of the various trials, with the exception of actinomycin D, whose dosage almost doubled (Table 1). The dose of ifosfamide was repeatedly readjusted in different risk arms [32]. Prognosis and survival in etoposide-induced leukemia remain extremely poor [56]. Etoposide was introduced in the high-risk arm of the EICESS 92 trial and was used to treat all risk arms with reduced doses starting in the EURO E.W.I.N.G. 99 trial [61]. After the introduction of etoposide in the EICESS 92 trial, more (45.5%) hematologic SMNs were detected than in the CESS 81 (0%) and CESS 86 (26.7%) trials without etoposide. However, the change in SMNs type after inclusion of etoposide may be due to several factors, including a longer follow-up period in the CESS 81/86 trials compared with subsequent trials. Finally, not only the cumulative dose but rather a combined use of chemotherapeutic classes such as anthracyclines with topoisomerase inhibitors or anthracyclines with alkylating agents might be crucial for the risk of leukemia [17,18]. Bhatia et al. confirmed that the risk of AML and MDS increased during treatments with a combination of anthracyclines (doxorubicin) and alkylating agents (cyclophosphamide, ifosfamide), as well as an increased cumulative dose of ifosfamide, cyclophosphamide, and doxorubicin [18]. Although there are several case reports of the association between chemotherapy and SMNs, few studies such as Bhatia et al.’s study have demonstrated a significant association between SMNs and the use of the latter chemotherapeutic agents [18,62]. It is also known that the use of G-CSF in combination with etoposide increases the risk of secondary leukemia [57]. For EwS, G-CSF was first used in the Euro-E.W.I.N.G. 99 trial. Although we found an increase in hematologic SMNs after this time period, its efficacy as an adjuvant to chemotherapy remains uncertain.

With an improved survival, late effects and also SMNs come more and more into the focus of pediatric oncology research. It is uncertain whether SMNs are a matter of treatment or genetics. The role of tumor predisposition syndromes in both the development of EwS itself and the development of SMNs after EwS remains elusive; associations between tumor predisposition syndromes and EwS have not been described [63]. In our cohort, there was no case of documented cancer predisposition syndrome. However, it remains unclear whether patients with EwS are predisposed to SMNs and whether those at higher risk can be identified upfront.

## 5. Conclusions

In conclusion, SMNs in EwS after successful treatment of primary EwS are rare events, but their occurrence is highly associated with devastating morbidity and mortality and thus worse prognosis.

Because we found differences in the development of SMNs among different study protocols, the risk of SMNs may be influenced by the dose and type of systemic treatment and radiotherapy. The retrospective nature of the present study and the combinatorial use of similar chemotherapeutic agents in the different trial protocols make it difficult to clearly assess the risk of single or combinatorial administration of alkylating agents and anthracyclines. The small number of SMNs and retrospective analysis are important limitations. Furthermore, the small cohort sizes and the predefined population limit the comparability of the above studies [24,37]. Other limitations in our analysis resulted from the loss of follow-up, particularly due to the transition from pediatric to general oncologists. This gap in consistent medical care underscores the importance of continued oncologic care, such as in specialized facilities that focus on adolescents and young adults. Given the rarity of EwS, international collaboration is needed to develop surveillance strategies that both prevent loss to follow-up due to transitions in medical care from childhood to adolescence and extend the duration of follow-up in late adulthood to capture potential SMNs.

## Figures and Tables

**Figure 1 cancers-14-05920-f001:**
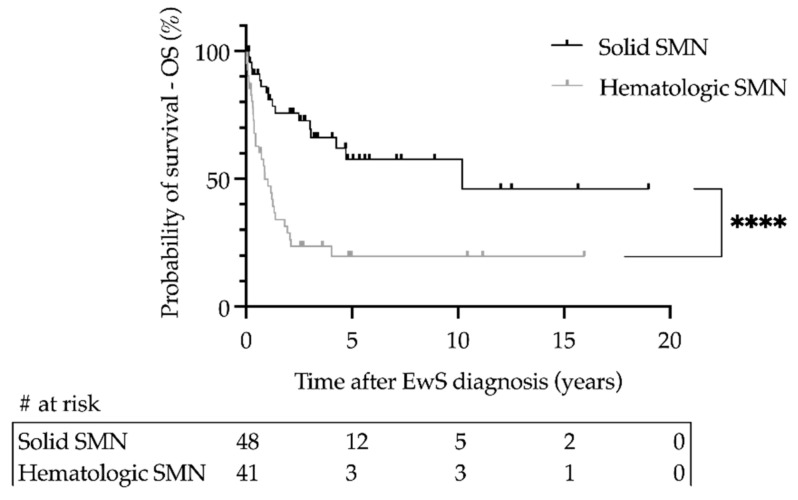
Survival of patients with primary Ewing sarcoma (EwS) and secondary malignant neoplasms (SMNs) as a function of secondary hematologic or solid malignancy in the CESS 81, CESS 86, EICESS 92, Euro-E.W.I.N.G. 99, and Ewing 2008 trials. Overall survival (OS) was calculated using the Kaplan–Meier method. **** *p* < 0.001. # at risk—number of patients at risk.

**Table 1 cancers-14-05920-t001:** Development of chemotherapy and radiotherapy for Ewing sarcoma (EwS) in different trials from the Cooperative Ewing Sarcoma Study (CESS) group.

EwS Trial		CESS 81	CESS 86		EICESS 92				EURO E.W.I.N.G. 99								Ewing 2008					
Number of Cycles		12	12		14				14		14		8	8			14		14	8	14	15
Risk Strata		-	SR	HR	SR		HR		R1 = SR		R2 = HR		R3 = VHR				R1 = SR		R2 = HR		R3 = VHR	
**Regimen**		VACA	VACA	VAIA	VAIA + VACA	VAIA	VAIA	EVAIA	VIDE + VAC ♀	VIDE + VAI ♂	VIDE + VAI	VIDE + VAI + BU/MEL	VIDE + VAI + ME/ME	VIDE + VAI + TREO/MEL	VIDE + VAI + BU/MEL	VIDE + VAI	VIDE + VAC ♀	VIDE + VAI ♂	VIDE + VAI	VIDE + VAI + BU/MEL	VIDE + VAC	VIDE + VAC + TREO/MEL
Chemotherapeutic Agent and Dose	V (mg/m^2^)	24	24	24	21				21			10.5	10.5				21			10.5	21	
	A (mg/m^2^)	6	6	6	10.5				12			1.5	1.5				12			1.5	12	
	C (g/m^2^)	14.4	14.4	-	12		-		10.5	-	-	-	-				12	-	-	-	12	
	I (g/m^2^)	-	-	72	24		84		60	102		60	60				54	102		60	54	
	D (mg/m^2^)	480	480	480	420				360				360				360					
	E (g/m^2^)	-	-	-	-			6.3	2.7				3.15	2.7			2.7					
	BU (mg/m^2^)	-	-	-	-				-			600	-	-	600	-	-	-	-	600	-	-
	MEL (mg/m^2^)	-	-	-	-				-			140	140			-	-	-	-	140	-	140
	TREO (g/m^2^)	-	-	-	-				-				-	36	-	-	-	-	-	-	-	36
Irradiation Dose (Gy)	Preoperative	-	-		45				54.4				54.4				54.4					
	Definitive	46–60	45–60 ↑		55				44.8–54.5 ↑				44.8–54.5 ↑				45–54 ↑					
	Postoperative	36	45		45				44.8–54.4				44.8–54.4				45–54					

SR—standard-risk, HR—high-risk, VHR—very high-risk. Please see section “Materials and Methods” for more detailed information on risk groups including randomization of R1/R2/R3. V—vincristine, A—actinomycin D, C—cyclophosphamide, I—ifosfamide, D—doxorubicin (adriamycin), E—etoposide, BU—busulfan, ME—melphalan, etoposide, MEL—melphalan, TREO—treosulfan. ↑ integrated boost to tumor target volume following predefined radiotherapeutic strategies within trial.

**Table 2 cancers-14-05920-t002:** Patient characteristics and clinical features of 96 primary Ewing sarcoma (EwS) patients with secondary malignant neoplasms (SMNs) in the CESS 81, CESS 86, EICESS 92, Euro-E.W.I.N.G. 99, and Ewing 2008 trials.

Attributable Distribution of Primary EwS Patients at Diagnosis	Number of Patients with SMNs (*n*, %)	Median Observation Time from Primary EwS Diagnosis to SMNs (Years)
**EwS trial (*n* = 96)**		
CESS 81	2 (of 184), 1.1%	21.7
CESS 86	16 (of 490), 3.3%	11.9
EICESS 92	21 (of 875), 2.4%	6
EURO E.W.I.N.G. 99	36 (of 1548), 2.3%	4.9
Ewing 2008	21 (of 1421), 1.5%	2.3
**Sex (%) (*n* = 96)**		
Male	45 (46.9%)	
Female	51 (53.1%)	
**Metastases (*n* = 96)**		
Yes	31 (32.3%)	
No	65 (67.7%)	
**Age (*n* = 96)**		
median (range)	14.4 (2.4–68.6) years	
**Localization (*n* = 96)**		
Cranium	5 (5.2%)	
Hand/foot	6 (6.3%)	
Upper limb	9 (9.4%)	
Lower limb	21 (21.9%)	
Axial skeleton	29 (30.2%)	
Pelvis	26 (27%)	

Clinical data of five patients with >1 subsequent malignant neoplasm are summarized in Appendix A.

**Table 3 cancers-14-05920-t003:** Types of secondary malignant neoplasms (SMNs) of 96 primary Ewing sarcoma (EwS) patients in the CESS 81, CESS 86, EICESS 92, Euro-E.W.I.N.G. 99, and Ewing 2008 trials.

EwS Trial	Type of SMNs	Number of Patients with SMNs (*n*, %)
Across trials (*n* = 96)	Solid	53 (55.2%)
	Hematologic	43 (44.8%)
CESS 81 (*n* = 2)	Solid	2 (100%)
	Osteosarcoma	0
	Other sarcoma	0
	Carcinoma	2
	Other	0
	Hematologic	0 (0%)
	Leukemia, lymphoma	0
	Myelodysplastic syndrome	0
CESS 86 (*n* = 16)	Solid	11 (73.3%)
	Osteosarcoma	4
	Other sarcoma	3
	Carcinoma	4
	Other	0
	Hematologic	4 (26.7%)
	Leukemia, lymphoma	3
	Myelodysplastic syndrome	1
EICESS 92 (*n* = 21)	Solid	12 (54.5%)
	Osteosarcoma	4
	Other sarcoma	2
	Carcinoma	5
	Other	1
	Hematologic	10 (45.5%)
	Leukemia, lymphoma	5
	Myelodysplastic syndrome	5
Euro E.W.I.N.G. 99 (*n* = 36)	Solid	18 (50%)
	Osteosarcoma	7
	Other sarcoma	2
	Carcinoma	7
	Other	2
	Hematologic	18 (50%)
	Leukemia, lymphoma	9
	Myelodysplastic syndrome	9
Ewing 2008 (*n* = 21)	Solid	9 (42.9%)
	Osteosarcoma	0
	Other sarcoma	2
	Carcinoma	5
	Other	2
	Hematologic	12 (57.1%)
	Leukemia, lymphoma	6
	Myelodysplastic syndrome	6

**Table 4 cancers-14-05920-t004:** Summary of data from secondary malignant neoplasms (SMNs) studies after primary tumors including Ewing sarcoma (EwS). AML—acute myeloid leukemia, BC—breast cancer, CI—cumulative incidence, CTX—chemotherapy, MDS—myelodysplastic syndrome, NA—not annotated; NMSC—Non-melanoma skin cancer, OS—osteosarcoma, RTX—radiotherapy, WLI—whole lung irradiation.

Study Details					EwS Characteristics at Diagnosis				Characteristics of Secondary Malignant Neoplasms						Comments
Author (Publication Year)	Reported Time Period (Time to Publication) (Years)	Cohort	Cohort Size (Patients)	Median Follow-Up (Range) (Years)	Median Age (Range) (Years)	Metastases (%)	Females (%)	Tumor Volume > 100 mL (%)	Number	Solid (%)	Predominant Type (%)	CI (%/Years)	Latency (Range) (Years)	Risk Factors	
Hawkins et al. (1996) [33]	1940–1983 (13)	EwS survivors	207	7.1	N A	N A	N A	N A	N A	N A	N A	5.4/20	N A	Sarcoma: CTX (alkylating agents, dose-dependent) Sarcoma: RTX (^4^/_5_ tumors in RTX field)	Selective description of secondary bone cancer after childhood cancer
Ginsberg et al. (2010) [34]	1970–1986 (24)	EwS	403	Alive: 23.0 (16–33) Deceased: 11.2 (5–28) *	13.5 (6–20)	N A	N A	N A	36	94.5	BC (36)	9/25	14.5 (4–32)	Solid: RTX (*p* = 0.28) BC: WLI	NMSC excluded
Kuttesch et al. (1996) [24]	1963–1990 (6)	EwS	266	9.5	14.2 (4.2–28; 90% < 21)	N A	56.25	N A	16	87.5	Sarcoma (62.5)	9.2/20	7.6 (3.5–25.7)	All SMN: RTX (>48 Gy) (*p* = 0.043) Sarcoma: RTX (100% in RTX field) (*p* = 0.002)	Combination of actinomycin D and RTX reduce risk
Dunst et al. (1998) [22]	1981–1991 (7)	EwS	674	5.1	13.25 (8–21)	25	87.5	50	8	37.5	AML (50)	4.7/15	6 (1.5–11.4)	Sarcoma: RTX (100% with RTX)	Selection bias for RTX
Navid et al. (2008) [35]	1979–2004 (4)	EwS family of tumors	237		N A	8.3	50	N A	12	33.3	MDS/leukemia (66.6)	4.7/10	3.3 (1.4–19.6)	Hematologic: CTX (alkylating agents, topoisomerase-II inhibitors, dose-dependent) All SMN: Localized stage (*p* = 0.036) Earlier treatment protocol (*p* = 0.001)	
Longhi et al. (2012) [36]	1983–2006 (6)	Localized EwS, <40 years	581	7.2	16.36 (6–39)	0	N A	N A	15	80	OS (40)	5/25	7 (1–21.1) Hematological: 3.1 Solid: 7.8	Female sex	
Friedmann et al. (2017) [37]	1974–2012 (5)	EwS, <40 years	300	7.8	MDS/AML: 17.4 (5–32) Solid: 14.6 (6–24)	30	30	30	15	60	MDS/AML (60)	15/25	10.9 (0.9–27.7) MDS/AML: 3.2 (0.9–4.6) Solid: 21.3 (10.5–27.7)	Hematologic: CTX (alkylating agents, topoisomerase-II inhibitors, dose-dependent)	NMSC and melanoma excluded

* Deceased patients within five years from EwS diagnosis excluded.

## Data Availability

The data presented in this study are available on request from the corresponding author.

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
