# Peer review of "Secondary Malignancies after Ewing Sarcoma—Epidemiological and Clinical Analysis of an International Trial Registry"

_cancers, 2022, doi:10.3390/cancers14235920_

Round 1

Reviewer 1 Report

This manuscript deals with a clinically very important issue. It is well structured/presented and involve; in contrasts to previously published and mostly older papers, a very high total number of patients followed for longer times than in the original clinical study reports.

Just a few aspects that need to be more clearly presented/discussed:

1. The statement that radiation doses above 60 Gy relates to the highest increse in secondary malignat neoplasms (SMNs) is unclear. In Table 1 none of the listed trials have standard given doses above 6o Gy.

2. The potential negative impact of different drug combinations (e.g. with/without actinomycin-D) should be more clearly presented/discussed

3. With the "current trend" to select EWS-patients with axially localized tumors just for definitive radiotherapy (in sted of surgery...to often with need for post-operative radiotherapy) - what could the (positive?) impact of current use of proton therapy be; with  considerably reduced low-dose-volumes of normal tissue(s) exposed to significant radiation doses.

4. Have the authors observed any trend related to SMNs in regard to fraction size of radiotherapy given?

Author Response

This manuscript deals with a clinically very important issue. It is well structured/presented and involve; in contrasts to previously published and mostly older papers, a very high total number of patients followed for longer times than in the original clinical study reports. Just a few aspects that need to be more clearly presented/discussed:

The statement that radiation doses above 60 Gy relates to the highest increase in secondary malignant neoplasms (SMNs) is unclear. In Table 1 none of the listed trials have standard given doses above 60 Gy.

Thank you for this important advice. The statement on that radiation doses above 60 Gy are associated with a higher SMN rate refers to a publication by Kuttesch et al. [24]. See line 348. It is correct that the protocols did not recommend doses above 60 Gy. The local therapy recommendation was not part of the randomized clinical trial and thus single patients were treated with higher dose when indicated by the caring physician.

The potential negative impact of different drug combinations (e.g. with/without actinomycin-D) should be more clearly presented/discussed.

Thank you for this important suggestion. You are absolutely right, and this would be a highly interesting question. We indeed have tried to analyze the impact of different agents on the frequency of SMN, however due to the heterogeneity of the protocols a solid and statistically meaningful analysis was not doable. 

With the "current trend" to select EWS-patients with axially localized tumors just for definitive radiotherapy (instead of surgery...to often with need for post-operative radiotherapy) - what could the (positive?) impact of current use of proton therapy be; with considerably reduced low-dose-volumes of normal tissue(s) exposed to significant radiation doses.

Thank you for your helpful comment and suggestion. In the literature there are indeed hints that unique dose-deposition pattern of proton therapy may be linked to a lower incidence of secondary malignancies compared to standard radiotherapy techniques. Rombi et al. described this positive impact in pediatric patients with Ewing sarcoma (Rombi B, DeLaney TF, MacDonald SM, Huang MS, Ebb DH, Liebsch NJ, et al. Proton radiotherapy for pediatric Ewing’s sarcoma: initial clinical outcomes. Int J Radiat Oncol Biol Phys (2012) 82(3):1142–8.10.1016/j.ijrobp.2011.03.038). In our investigation we decided not to analyze and compare the impact of proton therapy versus standard radiotherapy. In a broader range of patients in Europe was only used in recent years and was thus not analyzed in our cohort.

Have the authors observed any trend related to SMNs in regard to fraction size of radiotherapy given?

Unfortunately, the evaluation of the importance of different fraction types (conventional fractionation vs. hyperfractionation) was not possible as it was not documented in all cases especially in the early clinical trials. Furthermore, information like fraction size was not transferred to the database.

Reviewer 2 Report

The topic is very interesting and the study generally well designed. The paper is well organized and written.

The Authors reviewed consecutive Ewing sarcoma trials from the  to evaluate the features of secondary malignant tumors in EwS patients.

To draw any conclusion, the Authors should analyze the incidence of SMN in this population and to compare it to a similar population without EWS.

Osteosarcoma seems to have reduced over time?Might this be the consequence of amelioration in RTE regimen?

Is it a matter of treatment or genetics?The amelioration of survival is producing 2nd 3rd tumors in the same patient.

A plot reporting the cumulative risk of SMN over time would be an added value.

Author Response

The topic is very interesting and the study generally well designed. The paper is well organized and written.

The Authors reviewed consecutive Ewing sarcoma trials from the to evaluate the features of secondary malignant tumors in EwS patients.

To draw any conclusion, the Authors should analyze the incidence of SMN in this population and to compare it to a similar population without EWS.

Thank you for your feedback and advice. Our study was not designed as a case- control study but focuses on the outcome in Ewing Sarcoma. A detailed analysis using the data from all GPOH trials would be a different approach and would require data transfer agreements and ethics votes. Therefore, such an analysis is not doable in the near future. We however included in our discussion data from comparable analyses of other EwS therapy trials.

Osteosarcoma seems to have reduced over time? Might this be the consequence of amelioration in RTE regimen? 

Thank you for this helpful comment.  This is a meaningful hypothesis. In none of the trials, radiotherapy plans were collected and thus we cannot prove this hypothesis for our cohort. Given the improved radiotherapy techniques with IMRT, tomotherapy and proton beam radiotherapy it is of course possible that this may have impacted the decrease in Osteosarcoma.

We added this hypothesis to our discussion:

We hypothesize that the risk decreased over time due to the amelioration in radiotherapy regimen (for example the improvement of irradiation plans as well as techniques). In general, the initiation of preoperative radiotherapy in the EICESS 92 may have also opened the option of surgery (instead of definitive radiotherapy) for tumors deemed as unresectable before and therefore reduced the applied dosage.

Is it a matter of treatment or genetics? The amelioration of survival is producing 2nd 3rd tumors in the same patient.

Thank you for your helpful comment. Unfortunately, the cohort was not analyzed for tumor predisposition syndromes. And indeed, with an improved survival late effects and also second malignancies come more and more into the focus of pediatric oncology research. We added the question to our discussion.

A plot reporting the cumulative risk of SMN over time would be an added value.

We decided to use cumulative incidences as it was the best way to compare with yet reported results about our topic.

Reviewer 3 Report

This is a valuable article to read. It is very meaningful for clinicians to understand the secondary malignant tumor after treatment of Ewing sarcoma. However, the article has not clearly defined the concept of secondary malignant tumor after treatment of Ewing sarcoma, that is to say. The author should first tell the reader what is the secondary malignant tumor after treatment of Ewing sarcoma. In line 185, we see the shortest time of occurrence is 0.1 year after treatment. Whether the patient meets the diagnostic criteria here is questionable. Therefore, the author is requested to clarify the secondary malignant tumors after Ewing sarcoma treatment.

Author Response

This is a valuable article to read. It is very meaningful for clinicians to understand the secondary malignant tumor after treatment of Ewing sarcoma. However, the article has not clearly defined the concept of secondary malignant tumor after treatment of Ewing sarcoma, that is to say. The author should first tell the reader what the secondary malignant tumor after treatment of Ewing sarcoma is. In line 185, we see the shortest time of occurrence is 0.1 year after treatment. Whether the patient meets the diagnostic criteria here is questionable. Therefore, the author is requested to clarify the secondary malignant tumors after Ewing sarcoma treatment.

Thank you for your attentive input. Secondary malignancies are defined as tumors histologically distinct from a first treated malignancy as described in the introduction in line 69. We included all secondary malignancies occurring while or after the treatment and were not detected before the beginning of the first treatment. The mentioned case with only one month between first and second malignancy nevertheless suits to the definition as it presented as a histologically distinct tumor after a cancer treatment. Even if it matches with the definition, it is questionable if the second malignancy is related to the prior applied treatment or if it is a second primary malignancy besides the Ewing sarcoma.

We changed this to a more accurate definition in the introduction part of the paper: SMNs are defined as neoplasms that occur while or after cancer treatment and are not detected before the initial cancer treatment. Histologically SMNs present as distinct from the primary tumor.

Reviewer 4 Report

Can primary tumors be delineated by Ewing sarcoma vs. Non Ewing sarcoma? If so, how does this change the analysis of secondary malignancies?

It appears that the incidence of secondary malignancies increased with recent trials. With the description of prior clinical trials included in this report, please highlight the important differences that the authors hypothesize contribute to the incidence of SMN. Similarly, highlight these in Table 1.

SMNs after use of G-CSF has been controversial. Is there more recent supportive data that the authors could provide?

AEWS1031, the most recent children's oncology group trial for localized Ewing sarcoma reported higher early incidence of secondary malignancies. I would recommend that the authors reference this recent data in their manuscript and compare with their reported incidence. 

Author Response

Can primary tumors be delineated by Ewing sarcoma vs. Non- Ewing sarcoma? If so, how does this change the analysis of secondary malignancies?

Interesting point to analyze. The pathological methods were limited in the early study trials to a histological examination. Molecular analyses were not yet performed. Therefore, it was not possible to differ between Ewing sarcoma and Non- Ewing Sarcoma at this time. Pathological samples of the Euro Ewing 99 trial were examined retroactive with a methylome- classifier to differ between Ewing sarcoma and Non- Ewing sarcoma, but resulted in a minor number of Non- Ewing sarcoma. A sub analysis did not seem to be useful for this reason.

It appears that the incidence of secondary malignancies increased with recent trials. With the description of prior clinical trials included in this report, please highlight the important differences that the authors hypothesize contribute to the incidence of SMN. Similarly, highlight these in Table 1.

Thank you for this very helpful advice. The important differences that the authors hypothesize to contribute to the incidence of SMN are represented in table 3 as "risk factors". Also, we discussed the most important differences in the discussion field in several parts each focusing on a special hypothesize (e.g. radiotherapy) and described results of prior clinical trials.

SMNs after use of G-CSF has been controversial. Is there more recent supportive data that the authors could provide? 

In fact, G- CSF was administered from on the Euro- E.W.I.N.G. 99 trial. Although we found an increase in hematologic SMN after this time period, we do not hold enough data to provide an impactful statement as almost all patients received G-CSF.

AEWS1031, the most recent children's oncology group trial for localized Ewing sarcoma reported higher early incidence of secondary malignancies. I would recommend that the authors reference this recent data in their manuscript and compare with their reported incidence. 

We will add the following statement in the discussion part:

The most recent children’s oncology group trial for localized EwS (AEWS1031) randomly assigned patients to two different regimens. In regimen A patients received a standard five- drug interval compressed chemotherapy for 17 cycles whereas experimental therapy with five cycles of vincristine, topotecan and cyclophosphamide within the 17 cycles was administered to patients in regimen B.

A numeric rate of 4.3 % SMN was reported in a total of 626 patients. A cumulative incidence to compare with, was not described as well as any information about SMN characteristics (e.g. tumor entity). Leavey et al could not find a significant difference in SMN events between the randomized arms of their study. Therefore, they hypothesized that vincristine, topotecan and cyclophosphamide did not influence the risk of SMN.

Round 2

Reviewer 2 Report

The Authors made good efforts in the attempt to ameliorate their paper. 

In my opinion, it now merits publication on cancers.